# Effects of Defects on Masonry Confinement with Inorganic Matrix Composites

**DOI:** 10.3390/ma16134737

**Published:** 2023-06-30

**Authors:** Gian Piero Lignola, Gaetano Manfredi, Andrea Prota

**Affiliations:** Department of Structures for Engineering and Architecture, University of Naples “Federico II”, 80125 Napoli, Italy

**Keywords:** confinement applications, defects, experimental scatter, experimental validation, fabric-reinforced cementitious matrix (FRCM) composites, stress vs. strain relationship, theoretical incremental approach

## Abstract

Fabric-reinforced cementitious matrix (FRCM) composites are currently considered a suitable solution for strengthening existing structures. Confinement applications are still being investigated, since experimental programs showed significant scatter in the results and theoretical models are struggling to become established as a consequence. The main aim of this study is the identification of potential sources of scatter in the confinement efficiency of FRCM wrappings, in defects such as fiber slip within the matrix or imperfect straightening of fibers, or premature failure of fibers once exposed after complete matrix cracking. A theoretical incremental approach is proposed to simulate such effects. The approach is incremental, but not iterative, so that no convergence is required and the incremental step size has an impact only on the smoothness of the nonlinear theoretical stress vs. strain curves of the FRCM confined material, among other simulation results. Theoretical results are compared to experimental outcomes of previous tests. The main source of variability can be identified in the cited defects, and the approach can be considered satisfactory to simulate the effects of defects and the high scatter found in experimental results; however, further uncertainties in the behavior of materials can be included in future refinements of this study.

## 1. Introduction

Fabric-reinforced cementitious matrix (FRCM) composite systems evolved from ferrocement, replacing the metallic reinforcement with dry-fiber fabrics. They are based on the coupling of a structural reinforcing mesh (fabric) in an inorganic matrix. Inorganic matrices cannot fully impregnate individual fibers. The strands of the FRCM reinforcing mesh can be typically individually coated, but usually not fully impregnated; in fact, the term “dry fiber” is used to characterize an FRCM mesh, and this characterizes its behavior. The use of FRCM composites is currently acknowledged as a strengthening system able to enhance the load-carrying capacity of structural members [1,2,3,4]. They, at the same time, are able to guarantee mechanical and material compatibility, as it is required in retrofitting techniques for cultural heritage sites aimed at reducing invasiveness and ensuring reversibility and compatibility with the substrate [5,6]. Generally, a single ply of external strengthening yields a useful increase in bearing capacity, but it is potentially scattered. Conversely, strengthening systems applied in multi-ply strengthening schemes generate a significant increase in terms of strength and deformability, but they could hide some drawbacks [7,8].

The conservation and preservation of existing buildings, in particular, those classified as cultural heritage sites, are bringing together researchers worldwide [9,10]. The first step was the substitution of traditional techniques such as steel ties or reinforced concrete jackets with the first generation of fiber-reinforced materials with organic matrices, namely, Fiber-Reinforced Polymer (FRP) [11,12,13]. These have clear advantages such as high a strength-to-weight ratio, good durability, and the possibility of being tailored to structural requirements, but mainly because of the resin, they suffer drawbacks when applied to masonry structures, namely, poor composite–substrate compatibility, low permeability, and difficulties in removability [5,6,14,15,16]. To solve these problems, the organic matrix was replaced by an inorganic one [7,15,16]. In fact, focusing on masonry structures, the inorganic matrix has analogous behavior with the usual mortars in existing masonry structures, but the fibers have significant tensile strength [17,18]. At the same time, a reasonable level of removability, if not properly reversible, can be ensured. These new types of composites, FRCMs, have been used in different ways, while being mostly based on the same high-strength fiber grids and inorganic (mortar) matrix with different thicknesses and mechanical properties, namely lime-based, cement-based, and geopolymers. The most recurring names are Inorganic Matrix Grids (IMG), Textile Reinforced Mortar (TRM), or Steel Reinforced Grout (SRG) when particular fiber types are used (in that case steel cords).

All these combinations yield to different observed behaviors, and the research community decided to start numerous Round Robin Test (RRT) initiatives to deepen its knowledge and try to provide first guidelines and technical documents [19]. One of the first RRTs was promoted by Rilem Technical Committee 250-CSM (Composites for the Sustainable Strengthening of Masonry), who provided a general set of results on the tensile and bond behavior of several FRCMs considering various masonry substrates [20,21,22]. Another significant RRT of masonry columns made of clay bricks and Tuff stone and confined within a different number of layers of either glass FRCM or SRG was promoted by the Italian Department of Civil Protection with the involvement of eight Italian universities [23]. The outcomes focused on the influence of the number of layers of FRCM for different masonry types and FRCM strengthening systems, and the results were significantly scattered. This was a very systematic test program; however, other research has focused on the FRCM confinement of concrete and masonry columns. Such results show that numerous parameters alter the contribution of FRCM confinement to the axial strength of confined columns, namely, the mechanical and geometrical properties of substrate and composite, the number of fiber layers and matrix thickness, and their capacity [24,25,26]. The axial strength and deformability of confined columns depend not only on the number of layers, but also on the type of FRCM and the column cross-section aspect ratio [27,28,29]. However, very few or weak layers of FRCM may lead to minor increments of axial strength, even if the deformation capacity increases [15].

The matrix features play a major role; in fact, high-performance matrix can be similar to resin in FRP composites in protecting dry fibers and carrying the load, but if their stress contribution is very high, they can lead to brittle FRCM failure when they crack and the load is transferred to the fibers. On the other hand, low-quality matrix does not allow for the performance of the fibers to develop if they are prematurely exposed, and the stress is poorly transferred to the fibers when a crack opens within the matrix.

Some of the cited parameters have been the variables of experimental programs (e.g., column shape and performance of masonry; fibers, matrix behavior, strength, amount, to cite some examples [29,30,31,32,33,34,35,36]) and they have been included in available theoretical confinement models for confined members with FRP or FRCM (see [24,25,37] for a review). Unfortunately, such theoretical models are still not able to simulate all the experimental variability found in the cited programs. To understand the potential reasons for scatter in FRCM confinement performance is the main aim of this research, focusing on peculiar and potential defects in the confining system.

## 2. Research Significance

Results of experimental tests, even on nominally equal specimens, showed significant scatter, motivating the start of this research work. This scatter is mostly due to experimental material variability, however even with numerous specimens it can be found an unexpected outcome. For instance, the same brick masonry columns, strengthened with different number of wraps of the same jackets, show on average an increase in confined strength with the number of wraps, however single tested specimens with lower amount of wrapping can perform better than those with larger amount.

In accounting for potential variability in FRCM, it was already found that manual wet layup application can lead to scattered efficiency of strengthening systems due to intrinsic defects. These defects are the basis of this study and it is demonstrated that such defects can even reverse the relationship between confinement amount and confined material strength. It is also remarked that models dealing with ultimate stress of confining FRCM materials are not able to identify such variability, since defects do not necessarily lead to reduced FRCM material strengths, but they can impair FRCM behavior, hence the confinement follows different paths and only models involving the confinement evolution can trace the different performance.

It Is worth noting that design guidelines at present consider both FRP and FRCM with a linear behavior; hence, they implicitly assume the matrix is already cracked at peak load. This is not very important for models directly accounting for ultimate conditions (i.e., confined material strength only) or for safe design conditions, however it can be significant in the interpretation of the experimental behavior of tested specimens (where the matrix is uncracked at the beginning), or in the case of confinement models accounting for the entire stress vs. strain relationship of confined material.

## 3. Effect of Defects on FRCM Behavior

High-strength fibers embedded in an inorganic matrix are commonly used as a strengthening technique for existing structures, in particular for masonry, due to their low sensitivity to debonding between substrate and matrix.

However, the use of a lime- or cement-based mortar matrix instead of epoxy resin implies that attention should be paid to the behavior of the bond between the fibers and the matrix since cohesive slipping and straightening can occur in fibers in the mortar matrix.

The understanding of the mechanical properties of FRCMs, the modeling and design of appropriate experimental procedures for their mechanical characterization, and the modeling and formulation of appropriate design criteria are still discussed issues in the research community [19]. The main difficulty consists in managing their heterogeneous nature, which produces a strongly nonlinear mechanical behavior, making the interaction with the substrate on which they are applied even more complex.

The first reason is that FRCMs are mainly made by hand on site and applied directly to the element to be strengthened, with the fibers immersed in rather thick layers of mortar instead of thin resin. This can cause a more scattered behavior than FRP reinforcement systems because the linearity of the fibers and controlling the total thickness are more difficult.

Numerous theoretical and experimental studies have been conducted to characterize the mechanical behavior of FRCMs by investigating the local effects of the bond.

In particular, Bilotta and Lignola [38] theoretically analyzed the results of tensile tests performed in a round robin test on FRCMs with basalt fibers, glass fibers, PBOs, and aramid fibers. A great variability was found not only between the different laboratories, but also within each laboratory [39], highlighting the need to understand where this variability can derive from and possibly how to limit it to improve the characterization of FRCM systems [40,41].

Based on experimental results, De Felice et al. [19] proposed that the traction response of FRCMs can be divided into three stages, as shown in Figure 1. FRCM behavior has a linear relationship up to the peak stress of the first material reaching its linear elastic peak (usually the matrix), the so-called “uncracked state”. After matrix cracking, there is cracking (or softening) in the mortar; hence, its contribution reduces and cracks develop in the so-called “crack-developing stage”. Afterward, in the third stage, the so-called “crack-stabilized stage”, occurs when matrix contribution is almost lost. Since fibers are not protected by the matrix, this stage may be missing, depending on the combinations of properties in the matrix and fibers; this depicted as a dashed line and tends to overlap with dry fiber behavior (dotted line).

### 3.1. Matrix Behavior

The behavior of the matrix in tension is usually provided by a bilinear relationship with softening after peak stress, *f_m_*, at *ε_mu_* strain, and it drops to zero stress at an ultimate strain equal to *α*·*ε_mu_* (see Figure 2a), where *α* > 1 depends on the fracture energy properties of the mortar [8], which can be as large as 10 or even more (based on a smeared cracking approach [42]). After such a strain level, the matrix is fully cracked, exposing the fibers, whose behavior can be very scattered since the fibers are able to stretch further up to their ultimate strain or the system fails prematurely (i.e., the missing third stage). The matrix behavior is expressed by the following relationships (the tensile stresses, *σ_m_*, and strains, *ε_m_*, are positive):(1)$$\begin{array}{cc} \sigma_{m}=E_{m}\cdot \varepsilon_{m} & if\;0\leq \varepsilon_{m}\leq \varepsilon_{mu}\\ \sigma_{m}=E_{m}\cdot \frac{{\alpha \cdot \varepsilon }_{mu}-\varepsilon_{m}}{\alpha -1} & if\;\varepsilon_{mu}\leq \varepsilon_{m}\leq {\alpha \cdot \varepsilon }_{mu} \end{array},$$
where *E_m_* = *f_m_*/*ε_mu_* is Young’s modulus of the matrix.

### 3.2. Dry Fiber Behavior

The behavior of the (dry) fibers in tension is usually provided by a linear relationship up to peak stress, *f_fu_*, at a corresponding *ε_fu_* strain with brittle failure (see Figure 2b). After the matrix is fully cracked, fibers can continue to carry loads or break, thus yielding to premature failure (see Figure 1 and Figure 3, where the dashed line represents the uncertain behavior after the matrix fully cracks). This variability is even more noticeable if the fiber has defects and the linear portion starts after nonlinear behavior up to a certain stress level (known as transition stress, *σ_f,def_*). These defects implicitly account for fiber slip or imperfect straightening. Figure 2b shows different levels of defects, i.e., different curves at different transition stresses; namely, there is no defect if *σ_f,def_* = 0 and a maximum defect if *σ_f,def_* = *f_fu_*. In mathematical terms, the stress vs. strain behavior, originally described by Bilotta and Lignola [38], was improved in this research as follows (the tensile stresses, *σ_f_*, and strains, *ε_f_*, are positive):(2)$$\begin{array}{cc} \sigma_{f}=\frac{E_{f}\cdot \varepsilon_{f}^{2}}{2\varepsilon_{f,def}} & if\;0\leq \varepsilon_{f}\leq \varepsilon_{f,def}\\ \sigma_{f}=E_{f}\cdot \frac{{2\varepsilon }_{f}-\varepsilon_{f,def}}{2} & if\;\varepsilon_{f,def}\leq \varepsilon_{f}\leq \varepsilon_{fu}+\frac{\varepsilon_{f,def}}{2} \end{array},$$
where *E_f_* = *f_fu_*/*ε_fu_* is Young’s modulus of the fiber, and the transition strain is *ε_f,def_* = 2 *σ_f,def_*/*E_f_* ≤ 2*ε_fu_*. In this way, the linear branch is the entire curve without defect (if *σ_f,def_* = 0), and it reduces to a point with a maximum defect (if *σ_f,def_* = *f_fu_*) when the curve is entirely nonlinear. It is worth noting that defects impact the deformability of the fiber (nonlinear behavior); hence, the fiber strength, *f_fu_*, is attained at a strain of *ε_fu_* + *ε_f,def_*/2, higher than the expected *ε_fu_* with linear behavior.

### 3.3. FRCM Behavior

The behavior of the FRCM system (a parallel system, that is, the sum of the fibers and matrix and thus with a thickness equal to the sum of the two) in tension is usually provided by a combination of the two behaviors but accounting for an equivalent system with the thickness of the fibers only. The behavior in compression is generally neglected. For this reason, the equivalent stress, *σ_FRCM_*, is related to the total force in the system (tensile stresses and strains, *ε_FRCM_*, are positive) as follows:(3)$$\sigma_{FRCM}\cdot t_{f}=\sigma_{f}\cdot t_{f}+\sigma_{m}\cdot t_{m}\overset{yields}{\to }\sigma_{FRCM}=\sigma_{f}+\sigma_{m}\frac{t_{m}}{t_{f}}$$

The strain in the fibers and matrix is the same (i.e., strain compatibility, *ε_FRCM_* = *ε_m_* = *ε_f_*, is assumed up to fiber failure in a smeared crack approach [42]). Since *t_m_* >> *t_f_*, where *t_m_* is the thickness of the matrix, and *t_f_* is the equivalent thickness of the dry fibers, the contribution of the matrix is usually significant.

It is important that the previously defined “crack-developing” stage is stable; hence, the load carried by the FRCM during mortar cracking (i.e., at *ε_mu_*, equal to the fiber and matrix strains) should not be larger than the load carried by the fibers alone; otherwise, the FRCM system has brittle softening, and a potential abrupt premature rapture may occur. This condition can be satisfied without defects if
(4)$$E_{f}\cdot \varepsilon_{mu}+f_{m}\frac{t_{m}}{t_{f}}\leq f_{fu}\overset{yields}{\to }f_{m}\leq f_{fu}\frac{E_{m}\cdot t_{f}}{E_{m}\cdot t_{m}+E_{f}\cdot t_{f}}$$

Preferably, this crack-developing stage should not soften, and it starts at *ε_mu_* and ends at fiber failure or at complete mortar softening—whichever comes first, i.e., min (*ε_fu_* + *ε_f,def_*/2; *α·ε_mu_*). The third stage, the so-called “crack-stabilized stage”, occurs only if fibers did not already break; hence, if (*ε_fu_* + *ε_f,def_*/2)> *α·ε_mu_*, it strictly follows the dry fiber curve, and it starts at *α·ε_mu_* and ends at fiber failure, *ε_fu_* + *ε_f,def_*/2. Since fibers are not protected by the matrix, this stage is uncertain, in particular, if fibers have significant defects.

Figure 3 shows the global behavior of FRCM systems, for example, accounting for a typical mortar matrix cracking at quite low levels with moderate softening (namely, thin lime mortar, as it is usually required in historical masonry made of bricks applied with lime mortar). Each curve has a different defect level, the dashed lines represent the uncertain third stage, and the dotted line is the behavior of the dry fiber.

## 4. Proposed Theoretical Approach

### 4.1. Stress vs. Strain Relationship with Confinement

The axial stress–strain model (that is, the *σ_z_* vs. *ε_z_* relationship; in this case, the compression is positive) for the confined column is based on the equations proposed by Popovics [43]:(5)$$\sigma_{z}=\frac{f_{cc}\cdot x\cdot r}{r-1+x^{r}}$$
(6a)$$x=\frac{\varepsilon_{z}}{\varepsilon_{cc}}$$
(6b)$$r=\frac{E_{co}}{E_{co}-\frac{f_{cc}}{\varepsilon_{cc}}}$$
(7)$$\frac{\varepsilon_{cc}}{\varepsilon_{co}}=5\frac{f_{cc}}{f_{co}}-4$$
where *ε_cc_* is the strain corresponding to confined material strength, *f_cc_*, and *f_co_* is the unconfined strength. Furthermore, *E_co_* is the initial Young’s modulus, *x* is a normalized axial strain, and *r* is a model parameter. The “Popovics curve” is intended as a stress–strain relationship at a constant confining pressure; hence, *f_cc_* and *ε_cc_* are fixed, as is common when confinement is provided by a yielded material (constant confining pressure). In this case, the FRCM confining pressure is not constant but nonlinear; hence, a common practice is to consider an incremental approach and account for each axial strain, *ε_z,I_*, for a compatible constant lateral pressure, *f_l,i_*, and a confined material strength, *f_cc,i_*, to trace the curve. In this way a “Popovics curve” at a given constant confining pressure can be drawn, and the only “true” stress point, *σ_z,i_*, can be determined corresponding to the actual strain, *ε_z,i_*. The procedure is repeated up to the failure of the confining device, i.e., following the stress vs. strain evolution of the confining device.

The key aspect of this study, in fact, is the ability to account for the actual FRCM behavior in the development of the passive confinement pressure applied to the confined column. The nonlinear behavior of the confining material is coupled to the nonlinear behavior of the confined column, sensitive to the lateral pressure. Stress, *σ_FRCM_*, is related to strain, *ε_FRCM_*, but because of the assumed perfect bond between the confined column and the FRCM material, the FRCM strain is equal to the lateral column strain; i.e., *ε_FRCM_* = *ε_l_*.

### 4.2. Lateral-to-Axial Strain Relationship in the Confined Material

The lateral-to-axial strain (i.e., *ε_FRCM_* = *ε_l_* vs. *ε_z_*) relationship in the confined material provides the fundamental correlation between the behavior of the column and the behavior of the FRCM wrap in passive-confinement modeling.

The approach is virtually the same for different confined materials, namely, concrete, masonry, and so on. In this research, a reliable model proposed by Teng et al. [44] for confined concrete is used, while future advancements could include the use of more effective relationships for different confined materials. The absolute value of the secant slope of the lateral-to-axial strain curve of the confined concrete, ν = −*ε_l_*/*ε_z_*, applicable to unconfined, actively, and passively confined concrete, has a nonlinear implicit form:(8)$$\frac{\varepsilon_{z}}{\varepsilon_{co}}=0.85\left(1+8\frac{f_{l}}{f_{co}}\right)\left[{\left(1+0.75\frac{\varepsilon_{l}}{\varepsilon_{co}}\right)}^{0.7}-e^{-7\frac{\varepsilon_{l}}{\varepsilon_{co}}}\right]$$

In Equation (8), *ε_co_* is the unconfined peak material strain; usually, it is set equal to 0.002. The lateral pressure, *f_l_*, is commonly derived as
(9)$$f_{l}=k_{H}\frac{2{\cdot \sigma }_{FRCM}\cdot t_{f}}{D}$$
where *D* is the representative dimension of the cross-section, namely, the diameter for circular cross-sections and the side or diagonal for rectangular cross-sections. The reduced efficiency of confinement in the case of discrete wrapping along the axis of the column is neglected in this study, while the reduced efficiency of confinement, depending on the cross-section shape, is considered. As is well known, *k_H_* accounts for the shape of the cross-section, being equal to one in the case of circular cross-sections and lower than one in the case of rectangular cross-sections (side dimensions (*b*, *h*) and rounded corner radius (*r_c_*)). The coefficient, *k_H_*, can be assumed [3] to be
(10)$$k_{H}=1-\frac{{\left(b-r_{c}\right)}^{2}+{\left(h-r_{c}\right)}^{2}}{3\cdot b\cdot h}$$

To clearly focus on the defects of FRCM, the authors preferred not to introduce more refined evaluations of lateral pressure in cases of rectangular cross-sections, e.g., those proposed by the authors [45]; however, future developments could include the refinement of this part of confinement modeling.

Similarly, the last component of confinement modeling is the relationship between lateral pressure, *f_l_*, and confined material strength, *f_cc_*. In this case, there are numerous proposals in the scientific literature [24,25,37]; however, since the validation of the proposed theoretical research will be performed on clay brick masonry, the authors used a previous model specifically developed by them, which was validated for clay brick masonry [46]. The specific confinement equation for clay brick masonry is as follows:(11)$$\frac{f_{cc}}{f_{co}}=-1.07+2.07\sqrt{1+7.56\frac{f_{l}}{f_{co}}}-2.00\frac{f_{l}}{f_{co}}$$

### 4.3. Incremental Approach

As previously stated, the stress–strain model proposed by Popovics [43] accounts for constant confinement, but in this case, the confinement pressure is nonlinear. For this reason, an incremental approach is proposed; however, given the implicit nature of the adopted equation (Equation (8)), the analysis is not based on the evolution of the axial strain, *ε_z_*, but it is pragmatically based on the evolution of the lateral strain, *ε_l_*, since it can easily be assumed that during monotonic axial load evolution, the lateral strain increases too, from zero up to the ultimate value, corresponding to the failure of the wrapping system. In fact, the perfect bond assumption means that the FRCM system also evolves from zero strain (passive application) up to its ultimate failure strain. The calculation uses strain increment steps, each named ith step.

At each step, i, a lateral strain, *ε_l,i_*, is considered from 0 up to the attainment of fiber strength, *f_fu_*, at a strain of *ε_fu_* + *ε_f,def_*/2 since the lateral strain is equal to the FRCM strain (and fiber and matrix strain). Stress vs. strain relationships for FRCM can be obtained with Equation (3); based on matrix and fiber behaviors, Equations (1) and (2) can evaluate the FRCM stress at each step: *σ_FRCM,i_*. In turn, Equation (9) can evaluate the lateral pressure to be inserted in Equation (11) to obtain the confined clay masonry strength, *f_cc,i_* (with other models, it is possible to obtain concrete or other confined material strengths).

A Popovics curve can be traced with Equations (5)–(7), but since the lateral pressure and confined strength are related to a particular lateral strain, *ε_l,i_*, the only “true” stress point, *σ_z,i_*, along that curve can be determined corresponding to the strain, *ε_z,i_*, compatible with the lateral strain, *ε_l,i_*, provided by Equation (8).

This procedure is straightforward since there is no need for iterations or convergences. It can easily be incrementally repeated up to the failure of the FRCM system or earlier at a significant load drop on the softening branch of the confined column’s stress vs. strain relationship. As will be seen later, the usual experimental tests are stopped if there is a global load drop of 15 to 20% in the softening branch. This could not correspond to wrap failure but could occur earlier.

The entire aforementioned incremental approach is depicted in Figure 4.

## 5. Experimental Validation and Discussion of Proposed Approach

The capabilities of the proposed approach are shown with respect to the geometrical and mechanical properties of confined clay brick masonry columns wrapped in GFRCM composites, analyzed by Aiello et al. [23], who reported a recent round robin test at the Italian level. Despite nominally identical specimens being prepared and wrapped at the same laboratory and tested in different laboratories, variability in terms of the results was found not only between different laboratories but also in the outcomes of the same laboratory. This suggested that sources of such variability have to be found, and this represents a significant validation of the proposed approach in accounting for defects.

### 5.1. Outline of Past Experimental Program

The mechanical behavior of masonry columns confined by multi-ply FRCM systems was tested, subjected to uniaxial compressive load. The University of Salento prepared eight clay brick masonry columns (two unconfined and six confined) that were tested at the University of Naples. Compressive tests were carried out. Confined columns were strengthened by one, two, and three layers of GFRCM. The bricks had 125 × 250 × 55 mm^3^ dimensions, and the horizontal lime-based mortar joints’ thickness was 10 mm. Clay bricks had a tested compressive strength of 24.06 MPa, and for the mortar of joints, it was 4.35 MPa. Rounding of the corners (20 mm) was performed for each block with a computer-aided manufacturing tool in order to minimize human errors.

The columns had a square cross-section (250 × 250 mm^2^) and a length of 575 mm, and they were wrapped with a lime-based mortar and a dry glass mesh; the spacing of the mesh was 12 mm × 12 mm, with a 0.060 mm equivalent thickness in the two orthogonal directions and a density of 300 g/m^2^. The FRCM installation procedure was performed on soaked masonry, with two layers of mortar (5 mm thick each; 10 mm total), and a glass grid in between with an overlapping length of 250 mm according to Italian guideline CNR-DT215 [3]: a minimum overlap length equal to a quarter of the perimeter of the cross-section. In the cases of two and three mesh layers, the same application procedures were repeated, with a total thickness of 15 mm and 20 mm, respectively. An adhesion promoter, IPN-01, was used to protect the fibers from the alkaline environment of the mortar and improve the bond between the mesh and the mortar. To prevent any contact from the FRCM on the loading device and, hence, the direct axial loading of the jacket, a small portion at both the top and bottom, approximately 10 mm long, was left unconfined.

The laboratory at the University of Salento tested nine GFRCM specimens (dimensions, 600 × 60 × 10 mm^3^) equal to those used for the one-layer wrapping. Mechanical average properties with CoVs are reported in Table 1 and Figure 5 as they were tested at the University of Salento, where *E*_1_ and *E*_2_ refer to the slope of the first “uncracked” and second “crack-developing” stages (Figure 1); *f_fu_* and *ε_fu_* refer to peak strength and corresponding strain of FRCM, significantly lower than the dry fiber strength; *E_f_* is the dry-fiber Young’s modulus; and *f_mc_* is the mortar matrix’s compressive strength. Other properties of the FRCM materials (unavailable from the original tests)—namely, peak tensile stress, *f_m_* = 1.18 MPa, and Young’s modulus, *E_m_* = 2.4 GPa—for the mortar matrix were assumed via best fitting and were reasonably compared with the tested compressive strength, *f_mc_*, and a parameter of *α* = 10 was assumed.

Figure 5 shows the global behavior of the FRCM system; single crosses note the average tested behavior. It is clear that defects, as previously defined, also occurred in the direct tensile tests of the FRCM specimens since the post-cracking curve intersected with the elastic dry fiber ideal behavior (the black dotted line, no defect (*σ_f,def_* = 0), exactly simulates the experimental dry fiber behavior), and the theoretical model shows that the average FRCM experimental behavior reached the average defect theoretical curve with *σ_f,def_* = 50% *f_fu_*.

Finally, experimental outcomes for each of the eight unconfined and strengthened tested columns at the University of Naples are reported in Table 2. A hydraulic 3000 kN testing machine was used for displacement control with an external 1000 kN load cell. Four LVDTs, in both the vertical and horizontal directions, were used to assess axial and lateral strains. In Table 2, the values refer to in the first two URM rows (1_URM and 2_URM) are *f_co_*, *ε_co_*, and *E_co_*, while for the following confined columns, the first column is *f_cc_*. It is clear that the experimental increase in strength is not strictly proportional to the number of layers of FRCM; specifically, the two-layer FRCM columns (columns 5_RM_2L and 6_RM_2L) yielded to a lower masonry strength increase compared with the average masonry strength in the one-layer FRCM columns (columns 3_RM_1L and 4_RM_1L), while the worst column, with three layers of FRCM (7_RM_3L) had a lower strength than the highest masonry strength in the one-layer FRCM columns. These results can be partially explained by the typical experimental variability; however, other potential reasons are “defects”, as defined in this work. The next section shows the effect of the FRCM defects on the masonry’s confined strength, simulated with the proposed approach.

### 5.2. Discussion of Proposed Approach

To clarify the potential and features of the proposed approach, the entire incremental approach was applied to column 2_URM, one of the two unconfined columns with the lowest performance [23]. One layer of FRCM confinement was simulated with different levels of defects, as discussed in the previous section. The driving input parameter was the lateral strain, *ε_l_*, and in this case, increments of 0.1 mm/m were used. However, it is worth noting that a smaller increment size provides smoother curves, but the results are not sensitive to the step size.

The stress vs. strain curves for the masonry columns are based on equations proposed by Popovics [45]; the unconfined one is based on the experimental values in Table 2 for column 2_URM. The proposed incremental approach is applied, accounting for different levels of defects, and the dashed portion of the curves corresponds to the dashed portion of the FRCM behavior (i.e., Figure 5 and Figure 6), where the matrix is fully cracked and the third “crack-stabilized” stage occurs; hence, premature failure in the exposed dry fiber can be expected, which is a further source of uncertainty.

No defects or low defects (i.e., *σ_f,def_* = 25%*f_fu_*) yielded to monotonically increasing curves, and the confinement is effective; the dashed portion is quite small, and the increase in strength, *f_cc_*/*f_co_*, is up to 1.2 (only 1.12 if the matrix’s full cracking is assumed to be a safe threshold for FRCM effectiveness). Conversely, the higher the defect (i.e., *σ_f,def_* = 50%*f_fu_* or higher), the lower the confinement performance; in this case, the curve is not monotonic and the local peak and the following softening are mainly related to the softening behavior of the FRCM after matrix cracking. This occurs during the “crack-developing” stage of FRCM, and it yields to a local peak-confined column strength with a marginal increase in strength, *f_cc_*/*f_co_*, up to 1.04, almost independently of the defect level. Even if, in this case, only one layer of FRCM is considered, it is quite significant that the potential range of confinement effectiveness ranges from 1.04 up to 1.20 and, hence, 5 times the increment, depending only on the defects.

It is worth noting that, usually, experimental tests are stopped at a load drop after a peak of about 15 to 20%; hence, in some cases, the dashed hardening branch is not found experimentally because it could occur after a significant load drop,hence, after the experimental test is potentially stopped.

A confinement model not able to trace the full evolution of the confinement; namely, the entire stress vs. strain curve of the confined column is potentially not able to catch these peculiarities since the “defects”, as defined in this work, potentially do not alter the FRCM strength but rather how this strength is achieved. In other words, such models should focus on the endpoint of the confined curves in Figure 6, yielding almost the same confined strength, *f_cc_*. In fact, the higher the defect the higher the engagement of the lateral pressure on the confined column material’s softening. To better understand this aspect, a further outcome of the proposed approach is the evolution of lateral vs. axial strain, as depicted in Figure 7.

In accordance with the same format in Figure 5 and Figure 6, the ratio between lateral and axial strain is initially equal to the elastic Poisson ratio (about 0.25 on average), and while reaching the unconfined peak strain, the increasing cracking of the axially compressed column yields to proportionally higher ratios. Lateral strains and, hence, lateral pressure are usually modest at low axial strain levels before the dilation ratio starts to grow significantly, and this is the reason why the confined column curves are almost overlapping up to the unconfined peak strain. The dilation of column material is totally unrestricted in the case of unconfined material, but it is restrained by the confining wraps; in fact, at the same axial strain, the lateral strain is significantly smaller than the confinement. The higher the defect, the more the confinement effect occurs; hence, in the case of the maximum defect (i.e., *σ_f,def_* = *f_fu_*), the confined curve is very close to the unconfined one. However, the wraps are able to yield to higher axial and lateral strains compared with the unconfined column. The start in the third “crack-stabilized” stage, or, in other words, full matrix cracking, occurs at the same lateral strain, *α·ε_mu_*. However, different defect levels totally change the lateral pressure provided by the FRCM, hence the axial strain and corresponding stress in the confined column (see dashed portions in Figure 6 and Figure 7).

In conclusion, the proposed approach includes numerous effects and phenomena typically occurring in the development of confinement of a nonlinear material column with a nonlinear confining material, and in addition it includes the effects of defects in FRCM. In the next section, experimental validation is performed by extending this parametric evaluation, while also considering the experimental values in Table 2 for the other 1_URM unconfined tested column and assuming three values for the number of FRCM confining layers. The main outcomes are summarized in the following plots in Figure 8. The effect of the number of FRCM wrapping layers is given by the symbol in the curve, while the upper and lower curves with same symbols refer to the upper and lower expected performance of confinement (peak values of confined column stress) normalized to the considered unconfined column strength. Upper performance is measured assuming the attainment of fiber failure (i.e., *f_fu_* in the FRCM) while the lower performance is limited to full matrix cracking, i.e., the end of the second “crack-developing” stage in FRCM.

### 5.3. Experimental Validation

The earlier experimental program, previously recalled, is a meaningful case study to validate the proposed approach, since a comprehensive material characterization and specific parameters were considered in the research. Experimental results were significantly scattered, and the results in terms of each confined column strength were not proportional to the number of confining layers. Figure 8 clearly shows that the effect of the FRCM defects herein considered are able to generate high variability in the confinement effectiveness *f_cc_*/*f_co_* and the confined column strength with 3 layers, for example, can be lower than with one layer only, if defects are present (e.g., *σ_f,def_* = 25%*f_fu_* or higher).

Even without defects in FRCM, when focusing on the potential premature failure of the dry fiber exposed after full matrix cracking (i.e., in the third “crack stabilized” stage), the proposed approach is able to show that a 3-layer confinement could potentially lead to lower performance than a 2-layer confinement, and the same, respectively, with 2 and 1 layers. Figure 9, Figure 10 and Figure 11 show the comparison between each experimental axial stress vs. strain curve for masonry columns confined by one, two, or three layers of FRCM, respectively. The continuous line is the envelope of the different simulations. In each figure, not only is axial strain represented, but also the lateral strain. The black and red curves are the lower and upper bounds of the envelope of theoretical simulations; the dashed line is, as always, the portion corresponding to the third “crack-stabilized” stage and hence where premature exposed dry fiber failure can be expected. Furthermore, in the case of the upper bound, the analysis was extended up to dry fiber failure, since in this case FRCM failure occurred at a fiber stress quite a bit lower than the direct test result on dry fibers. For this reason, on the upper black curve, there is a cross (“X”) corresponding to the ultimate fiber strength, *f_fu_* = 891 MPa, for FRCM. In any case, the dashed line represents an uncertain portion of the curve that could go missing if premature dry exposed fiber failure occurs. The ability of the fibers to carry the load, once fully exposed by the cracked matrix, is considered more predictable if it is smoothly stretched, and this is more likely to occur in the case of lower numbers of layers.

Figure 9 shows the comparison with the masonry columns wrapped in 1 layer of FRCM: the two curves are significantly scattered. The experimental failure mode of the confined columns was reached, usually at the corner regions by fiber failure. In some cases, the failure was also associated with FRCM buckling (as bulging and debonding at mid-height) along the vertical direction, at lower loads (see [23] for photos of failed columns). This suggests that failure occurs after the matrix cracks and in some cases while FRCM is in the second “crack-developing” stage, hence prematurely when buckling occurred. Test 4_RM_1L has a high axial load capacity compatible with a low defect–high ultimate fiber strain condition.

Figure 10 shows the tested columns wrapped in 2 layers of FRCM, their performance is quite low and satisfactorily simulated by high defects in the FRCM. It is worth noting that the red dashed line is not strictly the lower bound because, if premature fiber failure occurs (as predicted in the third “crack stabilized” stage), there will be a potentially brittle load drop and the behavior depicted by the dashed line will go missing. At the same time, it is worth noting that lateral strain is experimentally measured over the mortar matrix by means of LVDTs, so that during crack failure those measures become unreliable.

Figure 11, finally, depicts the performance of columns tested with 3 layers of FRCM; their performance is quite high, hence the simulation yields a positive result with low defects in the FRCM. Both 2 and 3 layers are well simulated considering the failure at the end of the second “crack-developing” stage; hence, once the fibers are exposed, they tend to break prematurely at beginning of dashed line. One of the potential reasons for premature breakage can be the inclination of fibers, such as if they are not perfectly parallel (or if they are not perfectly orthogonal to column axis), hence there are some transverse components of force, jeopardizing the axial capacity of the dry fibers, and it is potentially more evident when more than one layer is applied. However, this is a preliminary conclusion, and further studies are required to fully comprehend the phenomena.

In conclusion, the validation can be considered satisfactory, since the main variability of the experimental results is confirmed by the proposed approach. It is worth noting that the main variability is due to defects in the FRCM, and variability in the unconfined masonry material is obtained by independently analyzing the two unconfined tested columns. Future developments could include a more comprehensive evaluation of uncertainties, since all the materials have scattered results (sometimes the CoVs of such properties are known, namely, masonry, mortar matrix, fibers); however, the focus of this research was to clarify the potential role of the “defects”, as they were considered herein, as a potential significant source of variability in confinement performance.

## 6. Conclusions

Confinement applications of FRCM materials are still under debate in the scientific community, since experimental programs show significant scatter in the results. The main aim of this study is the identification of potential sources of scatter in confinement efficiency of FRCM wrappings and find potential sources in the defects mainly due to hand application on site directly onto the element to be strengthened. These defects can be mainly fiber slip within the matrix, imperfect straightening of fibers, or premature failure of fibers once exposed after complete matrix cracking. These are peculiar to FRCM compared to other fiber-reinforced systems with organic matrices (i.e., FRPs).

A theoretical incremental approach has been proposed to simulate such effects. The simulations account for the full nonlinear behavior of both confined material and FRCM wraps. The approach is incremental, but not iterative, so that no convergence is required, and the incremental step size has an impact only on the smoothness of the theoretical curves. The simulation outcomes include the full stress vs. strain relationship of the FRCM-confined material and the confined vs. unconfined strength ratios. It is also remarked that models dealing with ultimate stress of confining FRCM materials are not able to identify such variability, since defects do not necessarily lead to a reduction in FRCM material strengths, but they can impair FRCM behavior; hence the confinement follows different paths and only models involving the confinement evolution can trace the different performance.

The effects of defects yield to ranges of mechanical responses, and the envelopes confirm that high scattering can be found. For instance, confinement effectiveness *f_cc_*/*f_co_* with three layers can be lower than with one layer only, if defects are present (e.g., *σ_f,def_* = 25%*f_fu_* or higher). Even without defects in FRCM, when focusing on the potential premature failure of the dry fibers exposed after full matrix cracking (i.e., in the third “crack-stabilized” stage), the proposed approach is able to show that a three-layer confinement could potentially lead to lower performance than a two-layer confinement. One of the other potential reasons for premature failure, not directly investigated here, might be the inclination of fibers, that is, if they are not perfectly parallel; hence there are some transverse components of force jeopardizing the axial capacity of the dry fibers, and this is potentially more evident when more than one layer is applied.

The proposed approach is able to simulate numerous effects and phenomena typically occurring in the development of confinement of a nonlinear material column with a nonlinear confining material, and in addition it includes the effects of defects in FRCM. Theoretical results are compared to experimental outcomes of previous tests conducted by the authors in a Round Robin test initiative. The main sources of variability can be identified in the cited defects, and the approach can be considered satisfactory in simulating the effects of defects and scatter found in experimental results; however, further uncertainties in the properties of all materials—namely, masonry, mortar matrix, fibers—and cross-section shape should be included in future refinements of this study.

## Figures and Tables

**Figure 1 materials-16-04737-f001:**
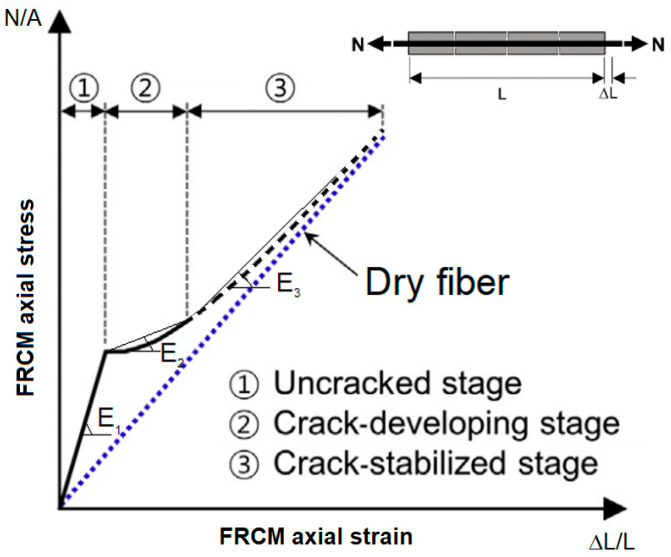
Tensile σ-ε relationship of an FRCM specimen.

**Figure 2 materials-16-04737-f002:**
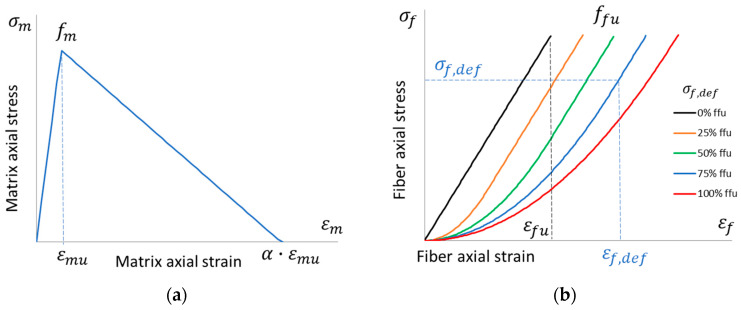
Stress vs. strain behavior in the tension of (**a**) the matrix, bilinear relationship; (**b**) dry fiber, nonlinear relationships with different defect levels: no defect, *σ_f,def_* = 0, up to a maximum defect if *σ_f,def_* = *f_fu_*.

**Figure 3 materials-16-04737-f003:**
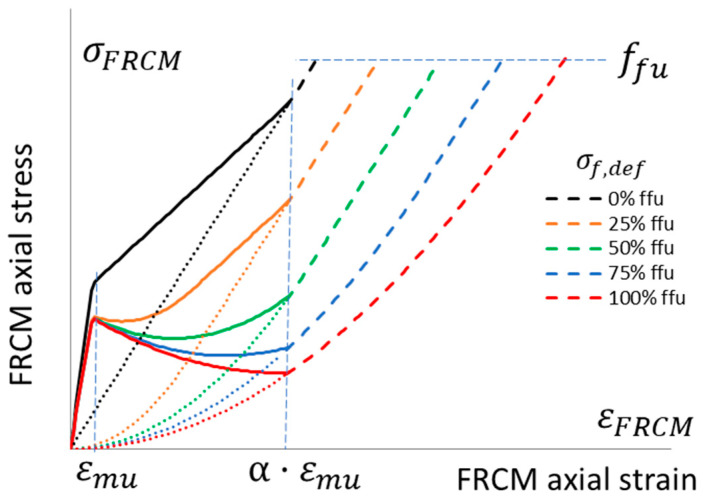
Stress vs. strain behavior in the tension of an FRCM system. Nonlinear relationships with different defect levels: no defect, *σ_f,def_* = 0, up to a maximum defect if *σ_f,def_* = *f_fu_*.

**Figure 4 materials-16-04737-f004:**
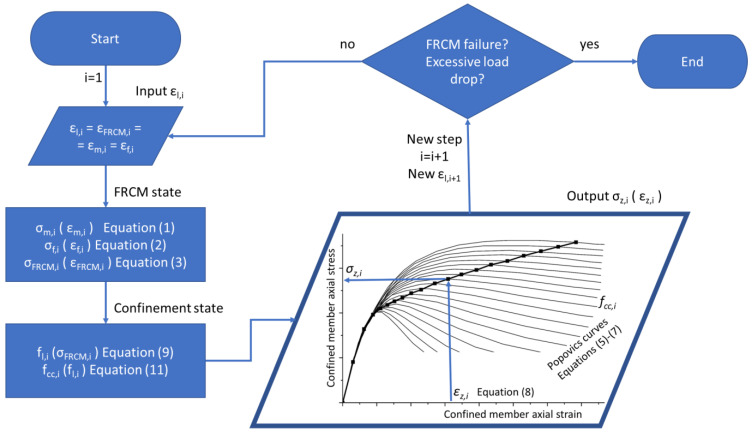
Flow chart of the proposed incremental approach.

**Figure 5 materials-16-04737-f005:**
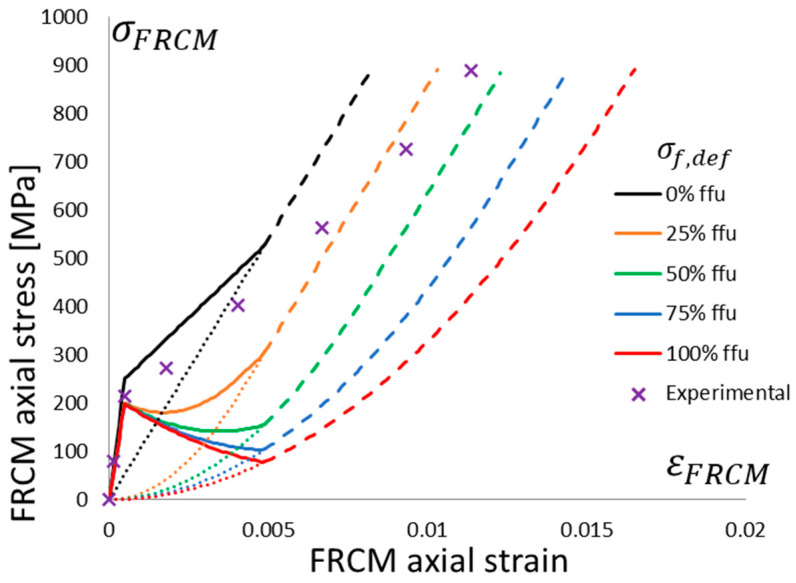
Stress vs. strain behavior in the tension of the tested FRCM system and theoretical nonlinear relationships with different defect levels: no defect, *σ_f,def_* = 0, up to a maximum defect if *σ_f,def_* = *f_fu_*.

**Figure 6 materials-16-04737-f006:**
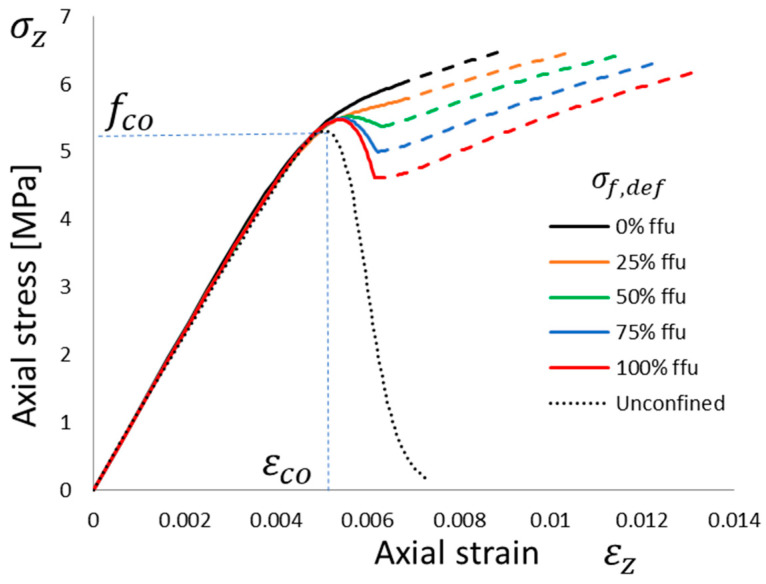
Theoretical stress vs. strain behavior of a column wrapped with one layer of FRCM with different defect levels: no defect, *σ_f,def_* = 0, up to maximum defect if *σ_f,def_* = *f_fu_*. Reference unconfined curve (simulating 2_URM) is visible.

**Figure 7 materials-16-04737-f007:**
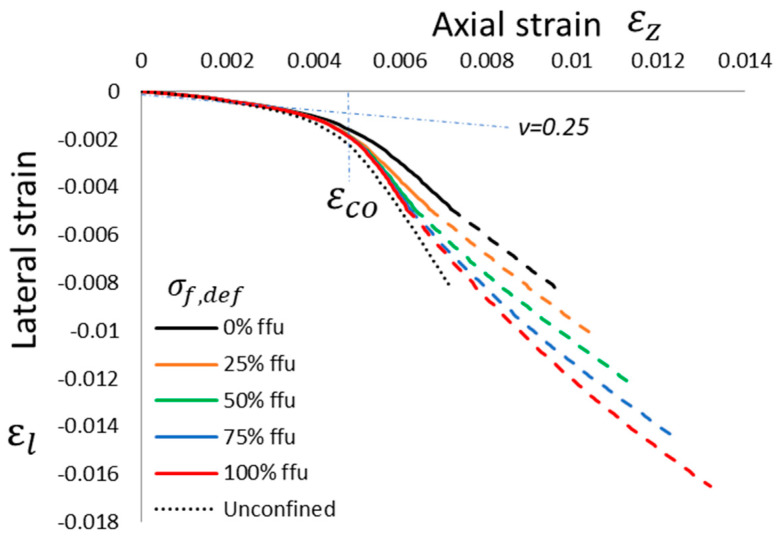
Theoretical lateral vs. axial strain behavior of a column wrapped with one layer of FRCM with different defect levels: no defect, *σ_f,def_* = 0, up to maximum defect if *σ_f,def_* = *f_fu_*. Reference unconfined curve (simulating 2_URM) is visible.

**Figure 8 materials-16-04737-f008:**
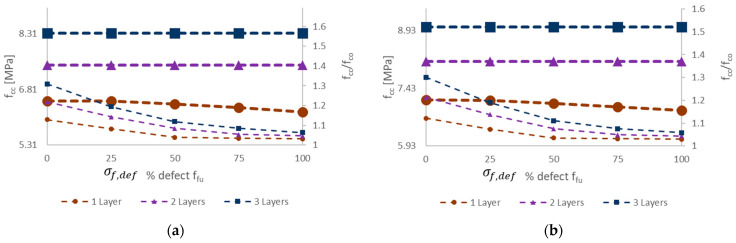
Confinement effectiveness including defects in FRCM wrapping (no defect *σ_f,def_* = 0, up to maximum defect if *σ_f,def_* = *f_fu_*) accounting for different number of layers, for clay brick masonry with: (**a**) *f_co_* = 5.31 MPa (i.e., 2_URM); (**b**) *f_co_* = 5.93 MPa (i.e., 1_URM).

**Figure 9 materials-16-04737-f009:**
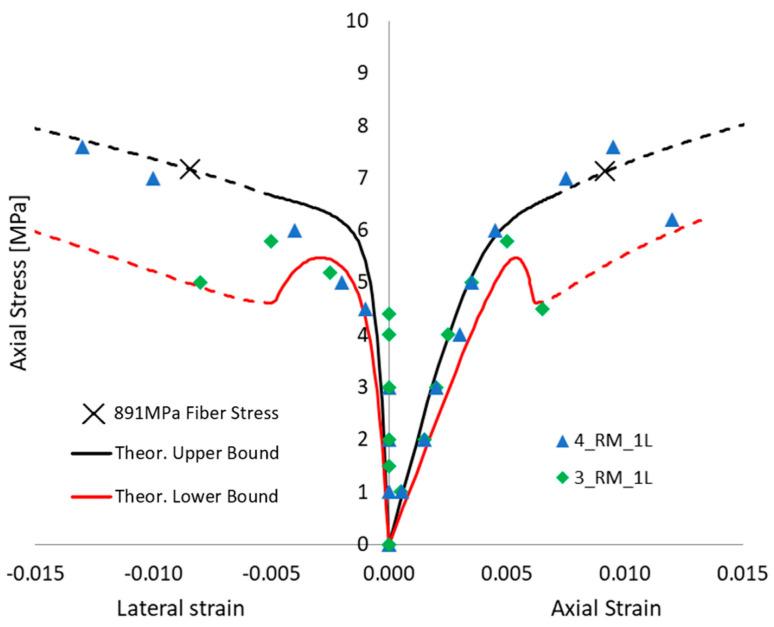
Axial stress vs. lateral and axial strain behavior of a column wrapped in one layer of FRCM. Theoretical envelope of the different simulations vs. experimental outcomes comparison.

**Figure 10 materials-16-04737-f010:**
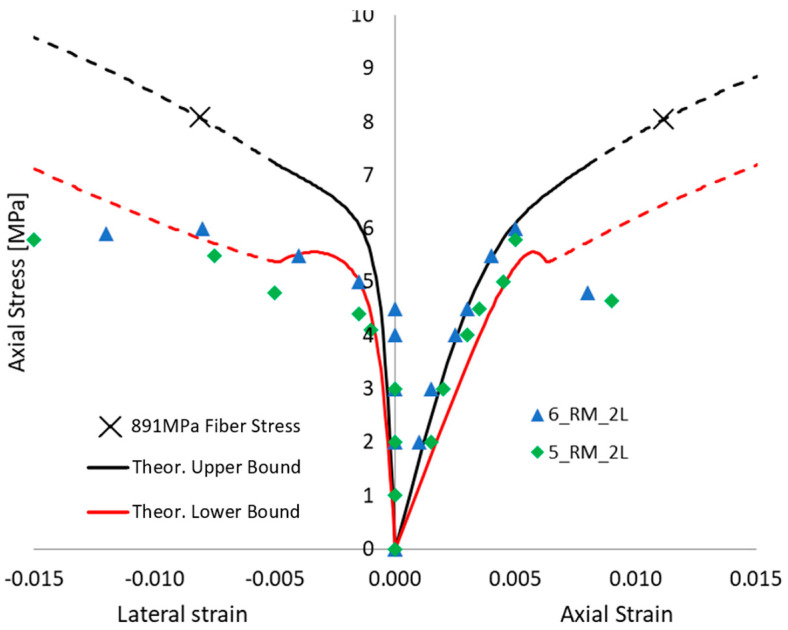
Axial stress vs. lateral and axial strain behavior of a column wrapped in two layers of FRCM. Theoretical envelope of the different simulations vs. experimental outcomes comparison.

**Figure 11 materials-16-04737-f011:**
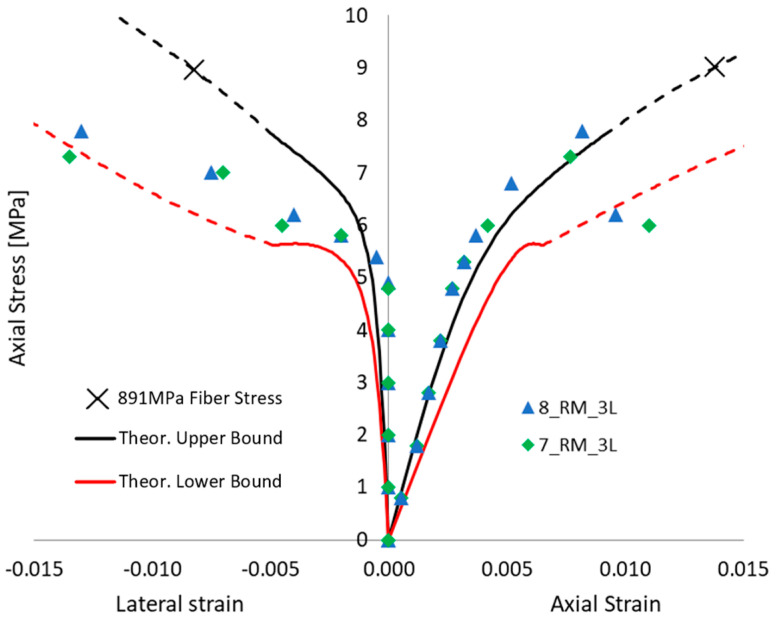
Axial stress vs. lateral and axial strain behavior of a column wrapped in three layers of FRCM. Theoretical envelope of the different simulations vs. experimental outcomes comparison.

**Table 1 materials-16-04737-t001:** FRCM mechanical average properties with CoVs, tested at the University of Salento.

Property	Mean Value	CoV
*E* _1_	514.5 GPa	0.08
*E* _2_	77.5 GPa	0.05
*f_fu_*	891 MPa	0.15
*ε_fu_*	9.7 mm/m	0.19
*E_f_*	108.0 GPa	0.16
*f_mc_*	9.1 MPa	0.27

**Table 2 materials-16-04737-t002:** Experimental unconfined and strengthened columns tested at the University of Naples.

ID	Max. Axial Stress (MPa)	Max. Axial Strain (mm/m)	Elastic Modulus (MPa)
1_URM	5.93	4.6	1674
2_URM	5.31	4.7	1173
3_RM_1L	5.57	4.1	2202
4_RM_1L	7.67	6.6	1858
5_RM_2L	5.66	5.5	2254
6_RM_2L	5.88	5.3	2755
7_RM_3L	7.39	7.7	2926
8_RM_3L	8.04	7.9	2669

## Data Availability

No new data were created in this study. Data sharing is not applicable to this article.

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
