# Peer review of "Effects of Defects on Masonry Confinement with Inorganic Matrix Composites"

_materials, 2023, doi:10.3390/ma16134737_

Round 1
Reviewer 1 Report
1. Author should clarify how he used highly nonlinear theoretical approach?
2. Author has fail to discuss on Fabric Reinforced Cementitious Matrix (FRCM) background, directly discussion on application , it is not acceptable.
3. I am surprised to see this statement " theoretical models are still not able to cover all the experimental variability".
4. Scope and objectives are not well written,
5. Research gap is missing
6. Author fails to discussion on experimental studies ( samples , methodology are missing).
7. I am surprised whether is experimental, numerical or review article. It is well defined.
8. Equation 6a & 6b should be separeted
9. Equations importance and description are missing
10. Figure 4 discussion is missing
11. Table 1, E2 values are showing less how this possible?
12. Conclusions should be in bullent point wise
13. Signifcance of research work is missing
Basic details should be clarified before proceeding further
Author Response
On behalf of all co-authors, I would like to thank the reviewer for her/his constructive comments. Providing an answer to these comments certainly increased the quality of the paper and made it more readable. The responses to the reviewer’s comments/queries are shown in the attached table. Column one shows the comments, while column two shows the responses and where in the paper the changes were incorporated

Reviewer 2 Report
Your study is very good, I can suggest to be published in this journal. Some corrections and clarifications needed:
1. Allowed self-citation is a maximum of 3 journals
2. Lines 63-68 should not be written in the introduction
3. Lines 82-86 should be given a literature review sub-chapter, supplemented by the research gaps that you are working on
4. Add an explanation of the methodology used separately from other chapters
5. Can you give an overview of the failures associated with line 471-475
6. An explanation of the second 'crack developing' stage should be provided in the discussion chapter
Minor editing of English language required
Author Response

(The authors gave the same response as above.)

Reviewer 3 Report
The paper presents a very interesting issue and proposes some explanation of the dispersion of results observed in laboratory tests. I agree that various types of defects in FRCM strengthening have a significant impact on the results of laboratory tests, and the qualitative and quantitative determination of this impact is very difficult.
The proposed approach allows for the correct recognition of the dispersion obtained in the citated study, which confirms the validity of the assumptions made and the impact of defects on the behavior of FRCM confinement columns.
The article is substantively correct, but I would ask for some clarification:
1) The authors enumerate factors that can be considered as appearing defects, however, it is puzzling - from a practical point of view - that a 'full defect' of FRCM occurs. In such a situation, it would have to be considered that the strengthening does not exist, so determining its impact is debatable. Please explain.
In practical terms, how should the authors' percentage treatment of the level of defect occurring be treated ?
2) Fig. 1 needs further clarification.
3) Please explain how (type of test) such characteristics were obtained for the matrix in tension. It is puzzling how these softening branches are determined in Fig. 2a
4) What is the basis for the determination of relationship 2?
5) 187-188 - No information is available on the extent to which strain compatibility εFRCM =εm =εf occurs?
6) 190-192 - Should the effect of mortar be included in the "crack developing stage" ? The authors write that "its contribution reduced”, but surely it should be analysed ? This is linked to the earlier question about the possibility of analysing (and determining) the softening branch in Fig. 2a. In general, when the matrix has been cracked it is not involved in the load transfer.
7) Figure 8 requires further explanation of the results obtained. It is puzzling that with 100% fibre failure no effect on performance of confinement was obtained (except for a slight reduction in case of 1 layer of FRCM) - please clarify.
8) The concordance obtained in Fig. 10 is not satisfactory - as the authors state - the results obtained in the study do not fit into the curves resulting from the proposed defect analyses.
The article also contains minor editorial incorrectness:
- the reference to fig. 2b is only below the figure, the same applies to Fig. 3 - please replace,
- often there is a lack of explanation of the designations in the equations - e.g.: tf and tm, they are obvious but need to be explained in the text; this also applies to other quantities used in the equations,
- too big drawings.
Correct language, allowing for easy understanding of the content.
Author Response

(The authors gave the same response as above.)

Reviewer 4 Report
Dear Authors,
thank you for a very interesting paper focused on the identification of potential sources of scatter in confinement efficiency of FRCM wrapping, in the defects like fiber slip within the matrix, or imperfect straightening of fibers. My comments are:
- line 193, there is "epsilon mu" - "mu" is a subscript (lower index)
- fig. 5, I would recommend using a brighter color for the crosses (experiment), gray is not very suitable, it is harder to see,
In this paper, a theory is presented that takes into account the effect of defects on FRCM behavior, which was also verified by experimental measurements (experimental samples). The paper is written clearly, and comprehensibly and I have no further comments.
Best regards.

Author Response

(The authors gave the same response as above.)

Round 2
Reviewer 1 Report
ok
ok